# Does eye-tracking have an effect on economic behavior?

**Jennifer Kee[1]☯, Melinda Knuth[2]☯, Joanna N. Lahey[3]☯, Marco A. Palma[1]☯ ***

**1** Department of Agricultural Economics, Texas A&M University, College Station, Texas, United States of America, **2** Department of Food and Resource Economics, University of Florida, Apopka, Florida, United States of America, **3** Bush School at Texas A&M University and NBER, College Station, Texas, United States of America

☯ These authors contributed equally to this work.
* mapalma@tamu.edu

**Data Availability Statement:** All relevant data are within the manuscript and its Supporting information files.

**Funding:** This project was funded by Texas A&M University. The funder had no role in study design,

## Abstract

Eye-tracking is becoming an increasingly popular tool for understanding the underlying behavior driving human decisions. However, an important unanswered methodological question is whether the use of an eye-tracking device itself induces changes in participants' behavior. We study this question using eight popular games in experimental economics chosen for their varying levels of theorized susceptibility to social desirability bias. We implement a simple between-subject design where participants are randomly assigned to either a control or an eye-tracking treatment. In seven of the eight games, eye-tracking did not produce different outcomes. In the Holt and Laury risk assessment (HL), subjects with multiple calibration attempts demonstrated more risk averse behavior in eye-tracking conditions. However, this effect only appeared during the first five (of ten) rounds. Because calibration difficulty is correlated with eye-tracking data quality, the standard practice of removing participants with low eye-tracking data quality resulted in no difference between the treatment and control groups in HL. Our results suggest that experiments may incorporate eye-tracking equipment without inducing changes in the economic behavior of participants, particularly after observations with low quality eye-tracking data are removed.

## Introduction

The application of eye-tracking technology to investigate human behavior in the social sciences has dramatically increased [1]. The main limitation to widespread use of such equipment has been its complexity and cost. However, eye-tracking technology has significantly improved over the last decade and is becoming more accessible as equipment prices continue to drop. Eye-tracking has proved to be a versatile methodological tool to evaluate intrinsic human motivations [2–10]. It has been used to understand choice processes in many fields, including psychology, neuroscience, computer science, economics, education, and other social and behavioral sciences; a web of science search shows that the number of articles with the keyword "eye-tracking" went from 71 in 2000, to 383 in 2010, and 1,919 in 2020. Within economic experiments, eye-tracking has been previously used to assess visual attention, pupil dilation, mental effort,

data collection and analysis, decision to publish, or
preparation of the manuscript.

**Competing interests:** The authors have declared
that no competing interests exist.

and, more recently, strategic interaction during economic games and to evaluate the nuances of behavior in game theory [9, 11]. In psychology and related fields it has been used extensively to study cognitive load, reading efficiency, complexity of information, attention, decision making under time pressure, and more [4, 6, 12, 13]. With the influx of experiments conducted with eye-tracking in the past decade, the use of eye-tracking will continue to gain popularity (see Lahey and Oxley [1] for a literature review, also Sickmann and Le [14] and Wang et al. [9]).

The beauty contest [15, 16], prisoner's dilemma [12, 17–20], leader games [20], public goods [21], two-armed bandit learning [22], variations of the normal form games of the prisoner's dilemma and stag hunt [3, 23–26], sender-receiver [9], coordination games [27, 28], risk preference [29–31], social preference [32], cheating game [33], and preference elicitation [34–39] have been studied using eye-tracking and other biometric equipment.

Even though eye-tracking provides rich data that enable researchers to better understand individual decisions, it is possible that participants may modify their behavior when they know their eye movements are being recorded [9]. Whether or not they do is an important methodological question given that most eye-tracking experiments are conducted in laboratory settings. Laboratory experiments have been subject of scrutiny and criticism as to whether they are representative of behavior outside of the laboratory. One of the most cited arguments against the generalizability of lab experiments is whether participants modify their behavior if they feel they are being observed [40]. This question has been previously investigated in a more generic domain without eye-tracking with the finding that participant's behavior is not affected by whether they are observed by the experimenters [41]. Although modern eye-tracking devices are highly unobtrusive (See Fig 1), Human Subject ethical standards often require consent forms to include information about the potential risks and a brief calibration stage at the beginning of the experiment may remind the participants that their eye movements are being recorded. (The risks associated with eye-tracking experiments are linked to participants with photosensitive epilepsy, cardiac pacemakers, and implantable cardioverter-defibrillators as the eye-tracker may interfere with them. Most of these participants are excluded from participation using pre-screening surveys). People may behave in a more socially desirable way if, by having their eyes monitored, they feel pressured to conform to social expectations [42]. Yet, the question of whether the use of an eye-tracking device itself induces a change in participants' behavior has not been comprehensively studied [9, 43], despite it being a common concern among grant and paper reviewers and seminar participants.

In this paper, we investigate whether the use of an eye-tracking device induces changes in the economic behavior in incentivized laboratory experiments. If it does, external validity would be limited to situations in which participants know they are being observed. We also study if any such effects go away over time. If they do, then later eye-tracking results can be used. We address these concerns using eight standard economic games, chosen for their varying levels of presence of socially desirable choices, from a standard double-auction game in which there is no socially preferred choice, to a cheating game in which the socially preferred choice is obvious. We randomly assign participants to either an eye-tracking treatment or a no eye-tracking control.

Social desirability is the tendency of some respondents to report an answer in a way they deem to be more socially acceptable [44, 45]. This behavior can be exhibited while engaging in social tasks through self-deceptive and impression management styles, causing measurement error [46, 47]. Social Desirability Bias (SDB) has been observed extensively in field and laboratory experiments [48–50]. Controlling for SDB can be difficult. Therefore, it is important to understand the experimental context in which SDB may arise [47, 51, 52].

Researchers from other fields have found evidence of brief-lived social desirability bias with eye-tracking technology using wearable eye-trackers or fake eye-trackers and provocative

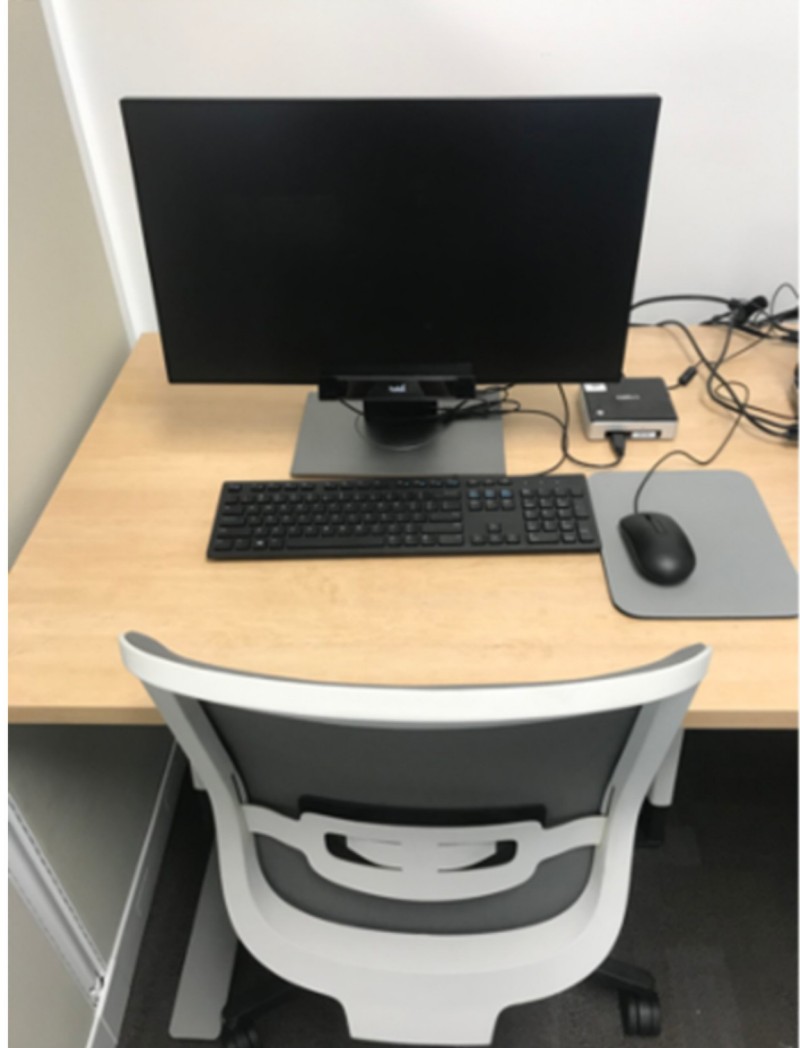

**Fig 1. A picture of a station set up within the lab.**

stimuli [53, 54]. These studies involved implicit deception, did not incentivize tasks monetarily, and made eye-trackers more salient than they would be in a standard economics experiment in which the eye-tracker is generally a small black box located at the base of a computer monitor (as shown for our experiment in Fig 1). Even with these differences, eye-tracking effects have been found to be short-lived, disappearing after 10 minutes without re-calibration [55].

Eye-tracking researchers sometimes check their own experiments for potential Hawthorne effects induced by eye-tracking. For example, Wang et al. [9] reports a null effect in a check comparing eye-tracking and non-eye-tracking participants in a sender-receiver game. Similarly, Harrison and Swarthout [56] had the same group of participants do their experiment with and without eye-tracking and found no difference in a risk lottery game. However, it is possible that there is publication bias for any individual experiment; papers not showing such an effect may be more likely to be published than those that do. Presently, to our knowledge, there is no study with the goal to comprehensively determine whether the physical presence

and explicit knowledge of eye-tracking equipment influences subjects' decisions in incentivized laboratory settings. The economic games for this experiment are canonical games with varied *ex ante* expectations depending on the game. The games used in this experiment include the Dictator game, Ultimatum game, Public Goods game, Trust game, Eckel and Grossman gambling risk task, Holt and Laury risk task, Double Auction, and a Cheating game. The null hypothesis is that there is no effect of eye-tracking on the economic behavior of participants for each economic game.

Overall, we do not find evidence of an eye-tracking effect in the economic behavior for seven of the eight games. Subjects neither behave more generously nor selfishly in the Dictator game, Ultimatum game, Public Goods game, and Trust game in the eye-tracking conditions. Meanwhile, they did not take less risk by choosing less risky gamble choices in the Eckel and Grossman gambling risk task, nor are they less likely to cheat by reporting a higher number in a Cheating game relative to the control condition. In the Double Auction task, in the raw data, there are no statistical differences in the average profits, transaction price, or bids or asks between the two groups, but there is a slight statistical difference in the transaction volume between the two groups. However, this difference disappears once the standard errors are clustered at the session level, and the indifference holds after controlling for the number of subjects and other demographic characteristics.

Only the Holt and Laury (HL) risk task showed a difference in means. Further exploration of these differences determined that in both the Holt and Laury risk task *and* the Eckel and Grossman risk task (using different sample populations), the number of calibration attempts was directly correlated with higher risk aversion. That is, for each additional calibration attempt, a participant's number of safe choices increased 0.2 points in both games. In the Holt and Laury game, these calibration outliers drove statistical differences between the treatment and control group. However, this effect was only present during the first five (of ten) rounds of the Holt and Laury game.

In eye-tracking research, it is common practice to remove participants who do not have eye-tracking accuracy above a certain "quality" threshold, such as 85% (Personal communication, iMotions, June 24, 2020.). After removing these observations from the treatment group, as would a researcher interested in the eye-tracking results themselves, the difference between the treatment and control groups in the Holt and Laury game is no longer significant. This result is likely because people who have difficulty calibrating also generate additional eye-tracking problems [58–60]. The question about the underlying path of the number of calibration attempts on risk behavior is interesting, but beyond the scope of this paper. Here, our main research question is whether eye-tracking causes a Hawthorne effect. Thus, for researchers who follow this standard practice of removing observations with poor eye-tracking data, it appears that even with a risk assessment task, a Hawthorne effect caused by use of the eye-tracker is not a problem in eye-tracking research using these standard economics games, that is, the act of being observed by the eye-tracker does not affect outcomes.

The rest of the paper includes the following sections: Section *Experimental Design* discusses the general experimental design, Section *Game Results* describes the setup and results of each game, Section *Number of Calibration Attempts and Risk Aversion* delves deeper into how calibration problems are correlated with risk aversion, and Section *Discussion and Conclusion* concludes this paper.

## Experimental design

A total of 404 students from Texas A&M University were recruited to participate over the course of 50 sessions ranging from 4 to 16 participants. Session times were available in the

morning and afternoon during regular university hours of 8:00 A.M.—5:00 P.M. Data were collected from July 2019 to February 2020. This study was approved by the Texas A&M University Institutional Review Board (IRB), protocol IRB2018–1602. The experiment was conducted at the Human Behavior Lab at Texas A&M University. The lab has 16 stations, each equipped with computers mounted with Tobii X2–60 eye-tracking devices and web cameras (Fig 1). Each station is surrounded by individual partitions six feet apart to prevent subjects from looking at other subjects. The experiment was computerized using Z-tree [61].

Tobii X2–60 eye-tracking devices are screen-mounted devices attached at the bottom of the computer monitor. The size of the eye-tracker is 18.4 cm length by 2.8 cm tall and 2.3 cm wide and is mounted on the computer monitor by a magnetic bracket. The device collects at the sampling rate of 60 Hz and is recommended for computers that have a 16:9 ratio (up to 25" screen). Subjects should be seated 40–90 cm from the eye-tracker and be situated so that their eyes are approximately at the same height as the middle of the screen for optimal operating precision (more detailed equipment specifics available at: https://www.tobiipro.com/siteassets/tobii-pro/user-manuals/tobii-pro-x2-60-eye-tracker-user-manual.pdf/?v=1.0.3). Subjects should remain in a fixed, comfortable position to prevent equipment failure by subject movement. Image sensors collect corneal reflection patterns as well as other visual data about the participant to document several eye metrics including saccades, fixations, pupil dilations, and visual pathways.

To be able to track eyes, the eye-tracking device must be calibrated to the eye movements of each subject. The calibration settings for this experiment included five calibration points (one in each of the four corners of the computer screen and one in the center). These points were in white with a light gray background. The calibration points moved across the screen in a left to right pattern and pulsed for a period of time in each of the corners. Subjects were to follow the dot with their eyes and could move their head naturally while facing forward towards the screen. Depending on the difficulty of the equipment to capture the pupil reflection pattern the process took approximately 15–30 seconds. Those who had difficulty could take up to 60 seconds to complete calibrate. Once the calibration was complete, a pop-up screen report was presented that indicated if the calibration was sufficient (was able to follow the pupil movements at least 80 percent of the calibration period) and to move on to the experiment or if the equipment needed to be re-calibrated (less than 80 percent) in which case the subject was re-positioned in front of the eye-tracking and the calibration process was re-initiated [61].

The experiment consisted of eight economic games, including five social interaction games: Dictator, Double Auction, Public Goods, Trust, and Ultimatum games; and three non-interactive games: Cheating game, and the Holt and Laury (HL), and Eckel and Grossman (EG) risk preference elicitation tasks. Conventional game instructions were used for all eight games [62–69]. Experimental instructions are available in S1 Appendix. Because of the large number of games, we will present the details for each game as well as predictions for social desirability bias along with its respective results in the results section.

We divided the eight games into two groups of four games to prevent subject fatigue. The games assigned to each group were selected to be balanced in terms of the game type and the amount of time required to complete the experiment. The games within Group 1 consisted of a one-shot Dictator game, Trust game, HL risk preference task, and a 10 period Double Auction. The games within Group 2 consist of the EG risk choice task, a 10 period Public Goods game, one-shot Ultimatum game, and 10 period Cheating game. The games for Group 1 were ordered from those theorized to have the most social desirability bias to the least, while the games for Group 2 were ordered from those theorized to have the least social desirability bias to the most. See Table 1 for details about the order of the games for each group.

**Table 1. Experimental sequence for Groups 1 and 2 including the number of sessions and observations between treatment conditions.**

| Conditions Calibration | Group 1 | | Group 2 | |
|---|---|---|---|---|
| | Treatment (Calibration) | Control (4.5 min. wait) | Treatment (Calibration) | Control (4.5 min. wait) |
| | Dictator | | Eckel and Grossman Gambling | |
| | Trust | | Public Goods | |
| | Holt and Laury Risk | | Ultimatum | |
| | Double Auction | | Cheating | |
| Number of Sessions | 14[A] (122) | 18[B] (126) | 8 (72) | 9 (76) |

Note. Observations are in parentheses. For Group 1, we have collected additional sessions, running only the HL Risk Task and Double Auction to reach the power.

[A] We collected 11(92).

[B] We collected 13(90) for Dictator and Trust game.

The treatment conditions and Groups were randomized at the session level. The experiment was conducted using a between-subject design with an *Eye-tracking treatment condition* and a *no eye-tracking control*. For the *no eye-tracking condition*, the eye-tracking device and video camera were turned off before the subjects entered the lab. The subjects were read a script with no verbiage about eye-tracking equipment or calibration. To balance the two conditions, they waited with a blank screen for 4.5 minutes, the average amount of time it takes for calibration in the *eye-tracking condition*. They were not forced to watch the screen during this time. Table 1 presents the number of subjects for each experimental condition. Group 1 has 100 more subjects than Group 2 because the Double Auction game in Group 1 requires larger number of subjects to play than do any of the games in Group 2. However, the number of subjects between the treatment and control conditions are balanced.

Subjects were asked to read and sign one of two consent forms depending on their treatment assignment. The consent form for the *eye-tracking condition* included specific language consenting to the use of eye-tracking during the experiment. The *no eye-tracking condition* consent form had no language about eye-tracking procedures and equipment. Once the consent form was signed, subjects were randomly seated at one of the eye-tracking stations. For the *eye-tracking condition*, the subjects were read a script indicating that they would be calibrated before the session began. Before starting each game, the subjects were verbally reminded about the eye-tracking equipment and re-calibrated. We intentionally re-calibrated after each game to remind the subjects of the presence of the equipment in the *eye-tracking condition* before each game, similar to Nasiopoulos et al. [55] so that the presence of the equipment would be salient at the beginning of each game.

To make the average expected payoffs similar across games, the exchange rate for each game varied from the original game play. Once all the subjects finished making decisions for all games and completed a demographic survey, the experimenter asked for a volunteer to draw a chip to determine the binding game and decision for payment. One of the four games (for each group) was randomly selected for payment then the subjects viewed their individual payoffs on their respective screens and received their payment in private. Subjects received a $10 show up fee plus the earnings they made based on their decisions in the binding game. Subjects were asked to sign a payment receipt form after they received their payment. The average payoff was $15.93 with a range of $14 to $20.

## Statistical power analysis

In order to determine the sample size requirements for each game, we conducted power analyses to obtain the optimal number of observations required for each game [63, 66, 68–75]. The

*a priori* power analysis required a minimum of 164 subjects for paired games: Dictator game, Trust game, Double Auction, and Ultimatum game, (82 per role) and a minimum of 82 subjects for the Public goods game and for the individual games: Holt and Laury, Eckel and Grossman, and Cheating game to have 80% power and a medium effect size (Cohen's D = 0.50) using G*Power. For paired games, we collected 182 subjects for Dictator and Trust games, 248 subjects for Double Auction, and 148 subjects for Ultimatum game. In individual games, we collected 248 subjects for Holt and Laury, and 148 subjects for Eckel and Grossman and the cheating game. The Public goods game, which consisted of groups of 4 and was played over 10 rounds, had 148 subjects.

In general, we ran more subjects for these games than did the papers that we cite, with the exception of Eckel et al. [71], which ran 245 subjects total, more than was needed for standard power analysis. More details, including the calculations and a list of the number of subjects from previous papers, are available from the authors upon request.

## Game results

Table 2 reports a test of whether the demographics, including gender, age in years, education, race, and income level, are balanced across the two conditions within group. The *eye-tracking condition* in Group 1 has older, more educated, but less-White individuals than does the *no eye-tracking control*. Additionally, the *eye-tracking condition* in Group 2 is younger and less educated than the *no eye-tracking control*. We control for these differences using regression

**Table 2. Demographic balance test across treatments.**

|  | Group 1 | | | Group 2 | | |
|---|---|---|---|---|---|---|
|  | No Eye tracking | Eye tracking | Kruskal-Wallis test | No Eye tracking | Eye tracking | Kruskal-Wallis test |
| Male[A] | 0.452 | 0.426 | P = 0.722 | 0.387 | 0.521 | P = 0.161 |
| Age (years) | 21.452 | 22.098 | P = 0.017 | 23.868 | 21.861 | P = 0.029 |
| *Education* |  |  | P = 0.013 |  |  | P = 0.038 |
| Freshman | 0.206 | 0.172 |  | 0.158 | 0.222 |  |
| Sophomore | 0.222 | 0.074 |  | 0.053 | 0.097 |  |
| Junior | 0.167 | 0.148 |  | 0.079 | 0.139 |  |
| Senior[+] | 0.167 | 0.221 |  | 0.197 | 0.208 |  |
| Master | 0.119 | 0.32 |  | 0.355 | 0.222 |  |
| Ph.D. | 0.119 | 0.066 |  | 0.158 | 0.111 |  |
| *Race* |  |  | P = 0.010 |  |  | P = 0.336 |
| White | 0.54 | 0.369 |  | 0.342 | 0.431 |  |
| Black | 0.056 | 0.066 |  | 0.013 | 0.028 |  |
| Asian | 0.31 | 0.402 |  | 0.526 | 0.431 |  |
| Others | 0.095 | 0.164 |  | 0.118 | 0.111 |  |
| *Income*[B] |  |  | P = 0.373 |  |  | P = 0.118 |
| <$45,000 | 0.389 | 0.375 |  | 0.473 | 0.375 |  |
| $45,000-$49,000 | 0.048 | 0.117 |  | 0.122 | 0.042 |  |
| $50,000-$59,000 | 0.063 | 0.125 |  | 0.054 | 0.083 |  |
| >$60,000 | 0.5 | 0.383 |  | 0.351 | 0.5 |  |
| N | 126 | 122 |  | 76 | 72 |  |

Note. Means and P-values from Kruskal-Wallis test are reported. Senior[+] includes both senior and 5[th] year in undergraduate.

[a] Group 2 had 75 observations for gender in the no eye-tracking condition and 71 observations for gender in the eye-tracking condition.

[b] Group 1 had 120 observations for income level in the eye-tracking condition. Group 2 had 74 observations for income level in the no eye-tracking condition.

analysis for each game. We present the results for each game in the following subsections based on *ex ante* expectations for each game and the order of play beginning with Group 1.

Each game was examined for differences in means and distributions using Mann-Whitney and Kolmogorov–Smirnov tests, respectively. Mann-Whitney and Kolmogorov-Smirnov tests estimate non-parametric differences in means and distributions between two independent groups. Note that using statistical tests to correct for false positives, such as the Bonferroni correction, would make our insignificant null result findings even less significant. As robustness checks for the mean comparison test results, we also report OLS regression in Table 3 including control variables. Randomizing treatments at the session level implies the possibility of correlations among subjects who participated in the same session. To account for this correlation, robust standard errors are clustered at the session level. Our results aligned with previous studies in Table D2 in S4 Appendix.

## Dictator game

The first game in Group 1 was the Dictator game, with instructions based on Andersen et al. [62]. In this game, subjects were matched randomly into pairs and were randomly assigned either the role of Player 1 or Player 2. Player 1 was asked to decide the number of tokens (out of 10) to split between themselves and Player 2. The exchange rate was 1 token equals \$1. Both Players 1 and 2 read through the instructions, then Player 1 made their decision while Player 2 was notified that Player 1 was making their choice. Subjects did not see the final result of the game unless it was chosen for payment at the end of the experiment. We hypothesized that if subjects feel the SDB in the *eye-tracking condition*, then they would transfer more tokens to Player 2 than in the *no eye-tracking control*. By doing this, they could be perceived as a generous person.

The giving behavior for Player 1 between the *eye-tracking* and *no eye-tracking conditions* is not significantly different. Fig 2 shows the comparison of the average amount of tokens sent by Player 1 between the conditions. There is no significant difference in the average number of tokens sent between the two conditions (MW, $p = 0.37$; regression, $\beta = -0.325$, $t = -0.66$). Like the mean comparison, we do not find differences in the distributions between the treatment and control (KS, $p = 0.866$).

We also estimate OLS regressions with demographic controls in Table 3, column (1) and find no change in sign or significance compared to the regressions without controls clustered at the session level. Additionally, the average number of tokens sent in both conditions, 2.77 tokens, is within the range that has been previously found in the literature [76].

## Trust game

Trust game instructions were based on Berg et al. [66]. Subjects were again paired with a random partner and randomly assigned either the role of Player 1 or Player 2. Both Players were endowed with 10 tokens. Player 1 was asked to decide the number of tokens to transfer to Player 2. Then, Player 2 received triple the number of tokens from Player 1. Player 2 selected the number of tokens to return to Player 1 from the available funds they had. The exchange rate was 1 token equals \$0.5. In this game, there is a chance that subjects might behave in a more socially desirable way in the *eye-tracking condition*. Player 1 in the *eye-tracking condition* might act more trusting by sending a higher number of tokens to Player 2 in the *no eye-tracking control* and Player 2 might return more tokens to Player 1 in the *eye-tracking condition* compared to the *no eye-tracking control*, showing higher levels of trustworthiness. Theoretically, since participants assume all players are being eye-tracked in the eye-tracking sessions,

**Table 3. Regression estimates for all games including demographic controls.**

| Variables | Coefficients (S.E.) | | | | | | | | | |
|---|---|---|---|---|---|---|---|---|---|---|
| | (1) Dictator game | (2) Trust game (Player 1) | (3) Trust game (Player 2) | (4) Holt and Laury | (5) Double Auction Profits | (6) Eckel and Grossman | (7) Public Goods | (8) Ultimatum game (Player 1) | (9) Ultimatum game (Player 2) | (10) Cheating game |
| Eye tracking | -0.254 | -0.053 | 0.01 | 0.545** | -0.165 | 0.125 | 3.283 | 0.475 | -0.096 | 0.136 |
| | (0.528) | (0.074) | (0.062) | (0.236) | (0.149) | (0.254) | (4.590) | (0.489) | (0.096) | (0.280) |
| Male | -0.363 | 0.051 | 0.018 | -0.693** | -0.137 | -0.348 | -12.515** | 0.354 | -0.016 | -0.388 |
| | (0.414) | (0.066) | (0.038) | (0.292) | (0.223) | (0.269) | (4.577) | (0.372) | (0.100) | (0.272) |
| Age | 0.143 | -0.014 | -0.010* | 0.097* | -0.053 | -0.065 | 1.347* | -0.012 | -0.015 | -0.031 |
| | (0.131) | (0.014) | (0.005) | (0.049) | (0.044) | (0.040) | (0.670) | (0.064) | (0.010) | (0.029) |
| Sophomore | 0.000 | -0.175 | 0.011 | -0.642 | 0.518 | -0.332 | 5.079 | 0.836 | 0.008 | -0.475 |
| | (0.802) | (0.124) | (0.073) | (0.431) | (0.340) | (0.355) | (11.129) | (0.723) | (0.132) | (0.428) |
| Junior | -1.270* | -0.113 | -0.068 | -0.566 | 0.326 | -0.87 | -4.562 | -0.236 | -0.174 | -0.098 |
| | (0.713) | (0.134) | (0.094) | (0.468) | (0.336) | (0.554) | (7.777) | (0.521) | (0.223) | (0.374) |
| Senior+ | -1.846** | -0.077 | -0.036 | -0.674 | 0.251 | -0.734*** | -3.098 | 0.768 | -0.009 | 0.492 |
| | (0.701) | (0.143) | (0.081) | (0.413) | (0.395) | (0.243) | (9.099) | (0.701) | (0.116) | (0.317) |
| Master | -1.826 | 0.046 | -0.012 | -0.858 | 0.568 | -0.307 | -7.937 | 1.448** | 0.075 | -0.217 |
| | (1.395) | (0.204) | (0.090) | (0.520) | (0.417) | (0.351) | (9.611) | (0.681) | (0.134) | (0.399) |
| Ph.D. | -1.709 | -0.021 | 0.015 | -0.917 | 1.053 | 0.219 | -19.318 | 0.908 | 0.002 | 0.335 |
| | (1.718) | (0.258) | (0.114) | (0.682) | (0.770) | (0.648) | (11.569) | (0.701) | (0.171) | (0.620) |
| Black | 2.321* | -0.021 | -0.140** | -0.572 | -1.029*** | -1.396** | 4.158 | -0.818 | 0.431* | 0.429 |
| | (1.227) | (0.134) | (0.066) | (0.690) | (0.287) | (0.616) | (8.131) | (0.546) | (0.242) | (0.499) |
| Asian | -0.787 | -0.209** | 0.016 | -0.162 | 0.023 | 0.339 | 8.736 | 0.448 | 0.098 | 0.775** |
| | (0.680) | (0.079) | (0.068) | (0.354) | (0.257) | (0.276) | (6.085) | (0.442) | (0.121) | (0.341) |
| Others | 0.342 | -0.082 | -0.039 | -0.365 | -0.236 | -0.483 | 0.221 | -0.297 | 0.175 | 0.389 |
| | (0.541) | (0.079) | (0.079) | (0.539) | (0.329) | (0.320) | (10.474) | (0.397) | (0.128) | (0.427) |
| $45,000-$49,000 | 0.341 | 0.338** | 0.196** | -0.431 | -0.204 | -0.75 | -2.436 | 1.028* | -0.233 | -0.326 |
| | (0.892) | (0.125) | (0.084) | (0.541) | (0.441) | (0.693) | (4.417) | (0.539) | (0.238) | (0.255) |
| $50,000-$59,000 | 0.37 | 0.059 | -0.11 | 0.238 | 0.143 | 0.286 | 4.85 | 0.277 | 0.170 | -0.386 |
| | (0.813) | (0.138) | (0.086) | (0.351) | (0.456) | (0.587) | (9.676) | (0.600) | (0.160) | (0.518) |
| >$60,000 | -0.371 | 0.044 | -0.031 | 0.142 | -0.190 | 0.14 | -8.978 | 0.97 | 0.122 | 0.231 |
| | (0.591) | (0.144) | (0.061) | (0.354) | (0.282) | (0.265) | (7.578) | (0.832) | (0.133) | (0.338) |
| Initial Offer | | | | | | | | | 0.479* | |
| | | | | | | | | | (0.040) | |
| Constant | 1.339 | 0.789** | 0.608*** | 3.352*** | 2.807*** | 6.328*** | 38.494** | 3.089 | 0.801* | 5.051*** |
| | (2.635) | (0.327) | (0.104) | (0.974) | (0.928) | (0.723) | (14.659) | (2.173) | (0.396) | (0.632) |
| N | 91 | 91 | 68 | 246 | 246 | 144 | 144 | 71 | 73 | 144 |
| R-squared | 0.205 | 0.211 | 0.205 | 0.08 | 0.063 | 0.147 | 0.156 | 0.222 | 0.278 | 0.104 |

Note. Coefficients from OLS estimations are reported. Standard errors are presented in parentheses and clustered at the session level. The dependent variables used in each game are: the number of tokens sent in the Dictator game; the proportion of the number of tokens sent is used for Player 1, whereas the reciprocity is used for Player 2 in Trust game; the number of safe choices in the Holt and Laury risk task; the average profits for 10 periods in the Double Auction; the number of safe choices in the Eckel and Grossman risk task; the average number of tokens kept in the Public Goods game; the number of tokens sent is used for Player 1, whereas the acceptance rate is used for Player 2 in the Ultimatum game; the average number of tokens reported for 10 periods in the Cheating game. For Holt and Laury, the results for multiple switchers are available from the authors upon request. A multiple switcher is someone who made multiple switches between Option A and Option B, demonstrating inconsistent behavior. An example of a multiple switcher would be someone who chose Option A until Decision 4 then switched to Option B in Decision 5 and switched back to Option A in Decision 6 and so on. It is common to report results with and without multiple switchers included [68, 83, 84]. Probit results with and without controlling for the initial amount of offer for the Ultimatum game are reported in Table D1 in S4 Appendix. Education is categorized as Freshman, Sophomore, Junior, Senior+, Master, and Ph.D. Race includes White, African American, Asian, and Others. Income is categorized as <$45,000, $45,000-$49,000, $50,000-$59,000, and >$60,000. Specifically, Senior+ includes both senior and 5th year in undergraduate. The results without controls are functionally equivalent and are available from the authors upon request.

* Statistically significant at 10% level;

** at 5% level;

*** at 1% level.

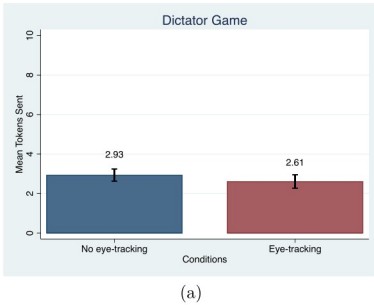
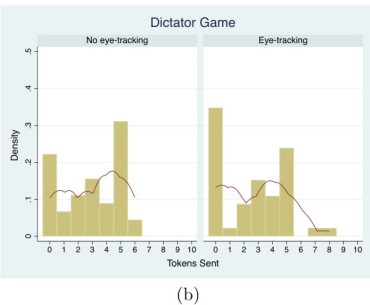

(a)                                                  (b)

**Fig 2. Mean and distribution comparisons in the number of tokens sent.** Note. Red lines in (b) present the univariate kernel density estimation.

eye-tracking sessions could have players assuming more generosity from their partners as well (see Frey and Bohnet [77]).

We do not find evidence of differences in behavior for either Players 1 or 2 across conditions. As a measure of trust, we used the fraction of tokens Player 1 sent to Player 2. As a measure of trustworthiness, we used the fraction of tokens Player 2 return. Fig 3 presents the mean difference in the proportions of tokens sent and returned between conditions. There is no difference in the proportion of tokens sent by Player 1 (MW, $p = 0.76$) and no difference in the proportion of tokens returned by Player 2 across conditions (MW, $p = 0.93$). Along with the mean comparisons, the distributions are not different between the *eye-tracking condition* and *no eye-tracking control* for either Player 1 (KS, $p = 0.78$; regression, $\beta = -0.037$, $t = -0.58$) or Player 2 (KS, $p = 1.00$; regression, $\beta = -0.013$, $t = -0.27$).

Results from OLS regressions controlling for different demographics, as provided in columns (2) and (3) of Table 3, are similarly insignificant for both Player 1 and Player 2 although the sign flips for Player 2. Eye-tracking does not affect behavior in the Trust game regardless of the inclusion of control variables. There is also no significant effect on trustworthiness in any regression results. In addition, the average proportion of tokens sent and returned in our study is consistent with previous studies [79, 80].

## Holt and Laury risk task

Subjects played the risk lottery game from Holt and Laury [68] (HL). This risk task consists of 10 Decisions with two safe and risky options labeled as A and B. As subjects progress through

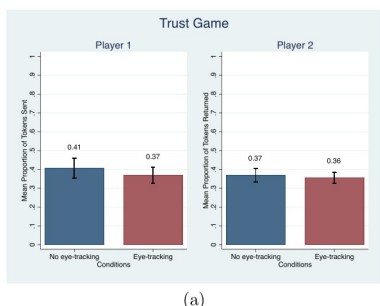
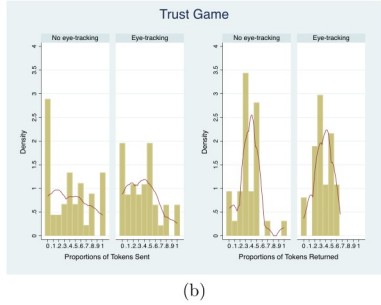

(a)                                                  (b)

**Fig 3. Mean and distribution comparisons in the proportions of tokens sent and returned.** Note. For reciprocity, those who received 0 tokens as Player 2 were treated as missing values [78]. The x-axis in the histogram represents the proportion of tokens the players received. Red lines in (b) present the univariate kernel density estimation.

**Table 4. Mann Whitney and Kolmogrov-Smirnov tests in early periods (1–5) and late periods (6–10) for the multiple periods games.**

| | (1) | (2) | (3) | (4) | (5) | (6) | (7) | (8) |
|---|---|---|---|---|---|---|---|---|
| | Holt and Laury | | Double Auction | | Public Goods game | | Cheating game | |
| | Early Periods | Late Periods | Early Periods | Late Periods | Early Periods | Late Periods | Early Periods | Late Periods |
| Eye-tracking | 4.107 | 0.918 | 1.641 | 1.628 | 57.978 | 63.397 | 4.244 | 5.372 |
| | (0.111) | (0.107) | (0.135) | (0.139) | (2.996) | (3.192) | (0.284) | (0.172) |
| No Eye-tracking | 3.754 | 0.770 | 1.851 | 1.830 | 57.389 | 66.205 | 4.095 | 5.279 |
| | (0.119) | (0.095) | (0.130) | (0.138) | (3.263) | (3.119) | (0.249) | (0.159) |
| P-value from MW test | 0.016 | 0.361 | 0.170 | 0.236 | 0.935 | 0.464 | 0.648 | 0.450 |
| P-value from KS test | 0.153 | 0.999 | 0.334 | 0.519 | 0.640 | 0.787 | 0.939 | 0.977 |
| N | 248 | 248 | 248 | 248 | 148 | 148 | 148 | 148 |

Note. Graphs and distributions are presented in Figs B1-B4 in S2 Appendix.

the Decisions, the probabilities for the payoffs in Option A and B change. Subjects are asked to choose one of the options in each Decision. Unlike the conventional HL game, in which all ten choices are displayed on a single screen, the game was modified so that each lottery decision was displayed on separate screens to ensure incentive compatibility [81]. The exchange rate was 1 token for $4. The subjects read through the instructions and went through one example. Then they participated in 10 real rounds in sequential order with a separate screen for each decision, meaning that the number of safe choices could vary between 0 and 10. We did not have strong predictions about the direction of socially desirable behavior; a person might prefer to choose more safe options in the *eye-tracking condition* than in the control in order to be perceived as a less risk-taking person. However, it is also possible that they might prefer to be perceived as more risk-taking and thus choose the opposite.

In contrast to the previous games, we observe an eye-tracking effect in HL. On average, subjects in both treatments showed risk averse behavior, meaning the average number of safe choices is greater than 4 [68], as can be seen in Fig 4(a). However, subjects in the *eye-tracking condition* have more risk averse behavior than those in the *no eye-tracking condition* (MW, $p = 0.04$; $\beta = 0.50$, $t = 2.19$). This difference also appears in the frequency distribution in Fig 4(b) (KS, $p = 0.07$). The distribution is slightly skewed to the right in the *no eye-tracking condition* compared to the *eye-tracking condition*. The sign and significance are the same for Poisson estimations (see Table D3 in S4 Appendix). However, the OLS is reported for ease of interpretation.

This difference between the treatment and control groups remains when demographic characteristics are controlled for ($\beta = 0.545$, $t = 2.31$), as shown in column (4) of Table 3.

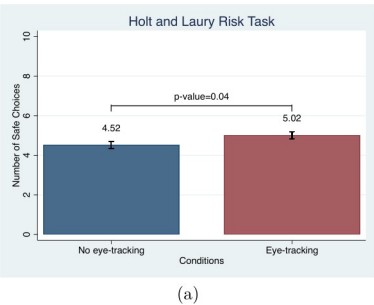

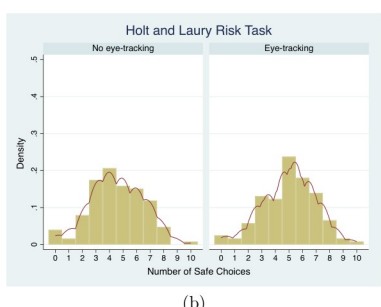

(a)  (b)

**Fig 4. Mean and distribution comparisons in the number of safe choices.** Note. Red lines in (b) present the univariate kernel density estimation.

(Consistent with the prior meta-analysis, females are significantly more risk averse than males [82]. This phenomenon is observed in both the *no eye-tracking condition* and *eye-tracking condition* (MW, $p = 0.03$ for the *no eye-tracking condition*; MW, $p = 0.07$ for the *eye-tracking condition*)). This difference still holds but loses significance for estimations that exclude participants showing inconsistent behavior, namely "multiple switchers," those who switch between Option A and Option B multiple times, ($\beta = 0.505$, $t = 1.86$ without controls; $\beta = 0.490$, $t = 1.73$ with demographic controls). In our study, the average number of safe choices for both conditions is 4.76. This number is similar to the original risk task study [68].

As discussed in the Introduction, it is possible that any eye-tracking effect is short-lived. We test this possibility by splitting HL into early rounds and late rounds and repeating the comparisons between eye-tracking and non-eyetracking group in Table 4, Columns (1) and (2). During the first five periods of the game, participants pick about 0.353 more risk averse choices in the eye-tracking condition compared to the no eye-tracking condition and this difference is significant at the 5 percent level. In the last five periods of the game, this difference drops to 0.15 and is no longer significant, suggesting that the eye-tracking effect does go away after several rounds of game-play. We further investigate the number of failed calibrations attempts taking eye-tracking and emotions data into account in section *Number of Calibration Attempts and Risk Aversion*.

## Double Auction

The last game in Group 1 was the Double Auction, with instructions based on Smith [69] and Attanasi et al. [65]. In this game, subjects were randomly assigned the role of either Buyer or Seller. The market consisted of 10 periods, each 2 minutes in length, and each subject had 1 unit of a fictitious good to buy or sell in each period depending on their role. The value for the Buyer varied from 1 to 8 and the cost for the Seller varied from 3 to 10. Buyers could not buy the goods at a price higher than their values and Sellers could not sell their goods at a price lower than their costs. Participants were informed that they could post their bids and asks in increments of 0.50 tokens. Before each period started, subjects were informed of their role and value or cost; these were fixed over the 10 periods and were displayed on the left side of the screen. Each period subjects could freely and simultaneously place a bid or ask in the market by inputting their bid or ask price at the bottom of the screen. If there was a price they wanted to buy or sell at, they could click the "Sell at this price" or "Buy at this price" button on the bottom of the screen. All transactions in the period were displayed on the right side of the screen. The remaining time was displayed on the top of the right side of the screen. The exchange rate was 1 token equals $1. We would expect no difference in the competitive equilibrium for the Double Auction game based on social desirability bias because behavior should follow basic economic theory and the parameters input in the game to maximize the utility of each player.

We use the average over the 10-periods of each of five dimensions to test for an eye-tracking effect in the Double Auction: profits, transaction price, transaction volume, bids, and asks. First, we examine the average profits over the 10 periods. We do not find any treatment effect. Fig 5 shows no differences in the mean and distribution of profits between conditions (MW, $p = 0.22$ and KS, $p = 0.39$). There is also no eye-tracking effect on the average transaction price (MW, $p = 0.84$ and KS, $p = 0.81$), average asks (MW, $p = 0.862$ and KS, $p = 0.973$), or average bids (MW, $p = 0.326$ and KS, $p = 0.612$). See the figures for these additional variables in Figs C1-C4 in S3 Appendix.

Differences in the mean transaction volume between the *eye-tracking* and *no eye-tracking conditions* are significant at the 10 percent level (MW, $p = 0.073$). However, once the standard errors are clustered at the session level using OLS, these effects are no longer statistically

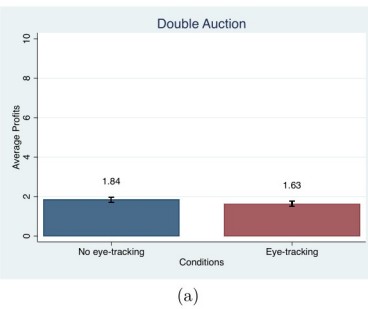
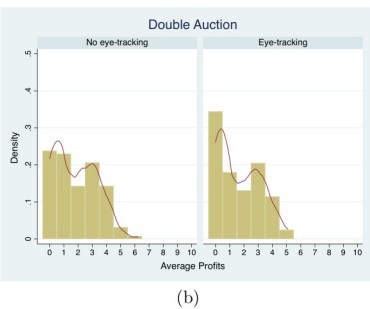

(a)  (b)

**Fig 5. Mean and distribution comparisons in the average profits of 10 periods.** Note. Red lines in (b) present the univariate kernel density estimation.

significant ($\beta = -0.064$, $t = -1.43$ for volume), nor are they significant when we control for the number of subjects ($\beta = -0.031$, $t = -1.17$). (The number of participants in a market should mechanically increase transaction volume, so to the extent that market sizes differ between the treatment and control groups, it should be controlled for when transaction volume is an outcome [85]). All auction results remain insignificant when demographic controls are added. With the inclusion of time trend for the periods, the results for the eye-tracking variable are similar and the time trend is significant for the average profits shown in Table D4 in S4 Appendix. We also analyzed the convergence to the equilibrium price and quantity along with the demand and supply; results (available from the authors) look as expected for double auctions, with no obvious differences between treatment and control groups.

### Eckel and Grossman Gambling Task

The first game in Group 2 was the Eckel and Grossman Gambling Task (EG), based on Eckel and Grossman [67]. Like the HL Risk Task, this task was a gamble choice task. Subjects were presented with six different gamble choices with two different payments in each choice and a 50–50 chance of each occurring. The alternatives increased in risk and expected return from Gamble 1 to Gamble 5. Gamble 6 had the same expected return as Gamble 5, but higher risk. Subjects were asked to choose only one gamble. This method of eliciting subject risk preferences has the advantage of requiring minimal math skills [67, 71, 86, 87]. The choice set in our investigation contained no loss of money (or tokens) as opposed to "loss" framework found in previous literature; the gambles were displayed as circles increasing in expected payoffs clockwise instead of in a table format. The exchange rate was 1 token equals $0.50. As with HL, subjects might pick more risk averse choices in the *eye-tracking condition* than subjects in the *no eye-tracking condition* or vice versa if they wish to be perceived as more or less risk seeking.

For EG, unlike HL, subjects' behavior is not significantly different in the *eye-tracking condition* compared to the *no eye-tracking condition*. The average mean gamble choice is measured to examine the treatment effect. Like the HL Risk Task, subjects reveal risk averse behavior by choosing Gamble Choices 1 and 2, as shown in Fig 6, which has been reverse coded (subtracted from 7) to show the mean safe gamble choice in order to make it comparable to HL. Again, unlike HL, there are no significant differences in the mean gamble choices or the distributions between conditions (MW, $p = 0.56$ and KS, $p = 1.00$).

Similarly, including demographic controls in an OLS framework provides no evidence of an eye-tracking effect, as demonstrated in column (6) of Table 3. This holds for the Poisson estimations (see Table D3 in S4 Appendix). The most frequently chosen gamble, 2 (reverse

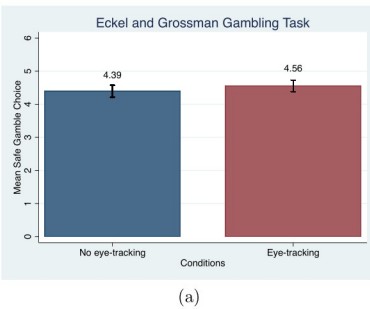
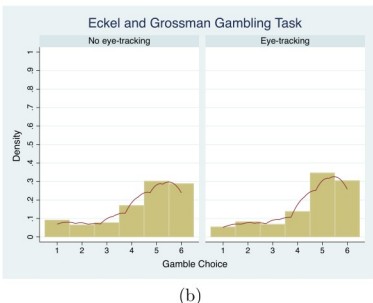

**Fig 6. Mean and distribution comparisons in the gamble choices.** Note. Red lines in (b) present the univariate kernel density estimation.

coded as 5 in our charts), and the average gamble choice, 2.53 (reverse coded as 4.47 in our charts), are close to that from previous studies, including one using the same sample population [67, 84].

## Public Goods game

The instructions for the Public Goods game were based on Andreoni [64]. Subjects were randomly assigned into groups of 4 each round and they played the game for 12 rounds (2 practice and 10 real rounds). In every period, each subject was endowed with 100 tokens. They were informed that they had two accounts, public and private, and were asked to allocate the number of tokens they wanted to go to each account. All tokens invested in the private account yielded a return of 10 cents. Each token invested into the public account, by all members, had a yield of 5 cents; all tokens invested to the Public account were doubled and distributed to all 4 members equally. At the end of each round, the participants viewed the number of tokens invested into their private account, the public account, and their earnings for that period. The rounds were not timed. The exchange rate was 1 token for $0.10. If there is social desirability bias, we would expect subjects to contribute a larger number of tokens to the public account in the *eye-tracking condition* compared to the *no eye-tracking condition* in order to be perceived as more cooperative.

Subjects in the *eye-tracking condition* do not behave differently than subjects in the *no eye-tracking condition*. The mean number of tokens kept in the private account is the metric used in this game. Fig 7 shows that no difference exists between the two conditions in the mean

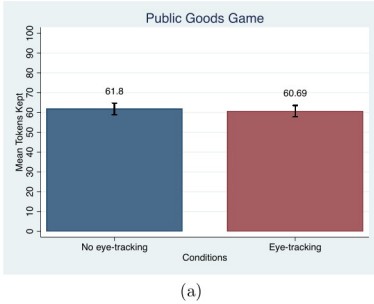
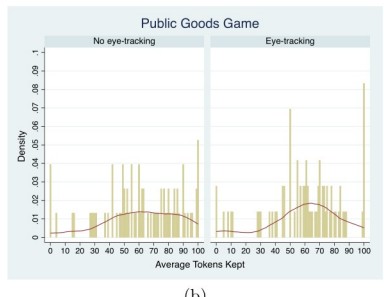

**Fig 7. Mean and distribution comparisons in the average number of tokens kept in 10 periods.** Note. Red lines in (b) present the univariate kernel density estimation.

tokens or the distribution of the tokens kept in the private account (MW, $p$ = 0.78 and KS, $p$ = 0.79). According to the distributions, allocating all tokens to the private account was chosen the most in both conditions. Keeping 50 tokens was the second most frequent choice. The distribution of tokens kept is centered around 65 in the *eye-tracking condition* whereas the tokens kept in the range from 30 to 100 is more evenly distributed for the *no eye-tracking condition*. Controlling for demographic characteristics does not change the lack of treatment effect, as shown in column (7) of Table 3. Although the time trend variable is significant, eye-tracking remains insignificant in Table D3 in S4 Appendix. The proportion of tokens kept in our study, 61.26%, is similar to that in a previous meta-analysis for the Public Goods game, but it is higher than that in other studies conducted at Texas A&M University [88, 89].

## Ultimatum game

Game instructions for the Ultimatum game were modified from Andersen et al. [63]. Subjects were randomly matched with a partner and were randomly assigned to be either Player 1 or Player 2. Similar to the Dictator game, Player 1 was endowed with 10 tokens and asked to select the number of tokens to transfer to Player 2. However, in this game, Player 2 could either reject or accept the offer from Player 1. If Player 2 rejected the offer, then both Players would receive nothing, otherwise the allocation was made according to the amount proposed by Player 1. The exchange rate was 1 token for $1. Social desirability bias would suggest that Player 1 would send more tokens in the *eye-tracking condition* compared to the *no eye-tracking condition*. The prediction for Player 2 is ambiguous.

Subjects again did not behave differently between the eye-tracking treatment and control in the Ultimatum game. We measured the mean number of tokens Player 1 sent and the acceptance rate of Player 2. Fig 8 shows the mean and distribution comparisons in the number of tokens sent and the acceptance rate between conditions. Both the average number of tokens sent by Player 1 and the acceptance rate of Player 2 are no different between conditions (MW, $p$ = 0.49 for Player 1 and MW, $p$ = 1.00 for Player 2). Similarly, the distributions are not different between conditions for both Player 1 and Player 2 (KS, $p$ = 0.99 for Player 1 and KS, $p$ = 1.00 for Player 2). OLS results available in columns (8) and (9) of Table 3 and probit results controlling for demographic characteristics in Table D1 in S4 Appendix. do not find evidence of an eye-tracking effect. The average endowment in the offers, 46.2%, and rejection rates, 16%, in our study align with a previous meta-analysis of the Ultimatum game [90, 91].

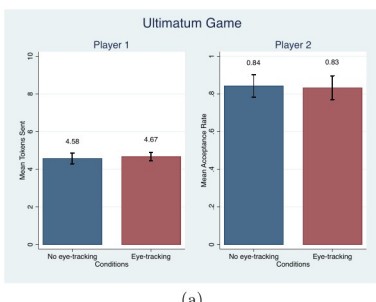
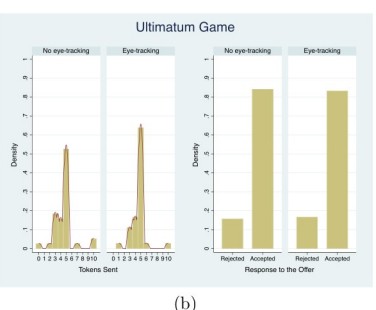

(a)　　　　　　　　　　　　　　　　　(b)

**Fig 8. Mean and distribution comparisons in the number of tokens sent and the acceptance of the offer.** Note. For player 2, acceptance is coded as 1 and rejection is coded as 0. Red lines in (b) present the univariate kernel density estimation.

## Cheating game

The final game in Group 2 was the Cheating game with instructions adapted from Aksoy and Palma [92], which is based on Fischbacher and Föllmi-Heusi [93]. This game involves 10 periods. In each period, subjects saw a random sequence of numbers consisting of 0, 2, 4, 6, and 8. They were asked to report the first number they saw in each period and were told that the number that they report was their payment. For example, if a participant saw the number 2 first (i.e. the true state of the world), but reported seeing the number 6, their payment was $6. Since the order of the numbers was randomly generated, and there was no enforcement or punishment for misreporting the number that participants actually saw first, they can lie by reporting a higher number than what they actually saw and profit from it. As promised, subjects were paid for the value that they reported. The exchange rate was 1 token equals $1. SDB would suggest that subjects would be less likely to lie, that is, the average reported number would be lower, in the *no eye-tracking condition* than in the *eye-tracking condition*.

In our results, eye-tracking does not affect the subjects' propensity to lie. The average number of tokens reported in 10 periods is used to measure the treatment effect in this game. The expected average number of tokens from the random number placement is 4 tokens (i.e. the expected value of the five numbers that appear on the computer screen in random order: 0, 2, 4, 6, 8). In our study, the average number of tokens reported in both conditions for the 10 periods is higher than the truth-telling expected number of tokens, about 4.75. Fig 9 shows the mean and distribution comparisons between conditions with the expected likelihood of 20% for each number under truth telling. There is no difference in the average number of tokens reported between conditions (MW, $p = 0.83$). The distributions for both conditions are slightly skewed to the right, showing signs of a small degree of lying, with no difference in the distribution (KS, $p = 0.98$). Again, controlling for demographic characteristics does not change the results, as shown in column (10) of Table 3. We also confirm that the average number of tokens reported in our study aligns with previous studies showing that there is a tendency for lying, but the magnitude is small [94].

## Number of calibration attempts and risk aversion

In the previous section, we show that there are no eye-tracking effects in the games except for the HL Risk Task, and this eye-tracking effect is only prevalent in the first half of the game. In this section, we dive deeper into the HL result and explore the effect of the number of calibration attempts on risk aversion in both the HL and EG risk tasks. We first show that the number

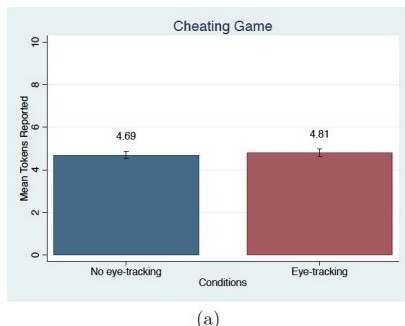
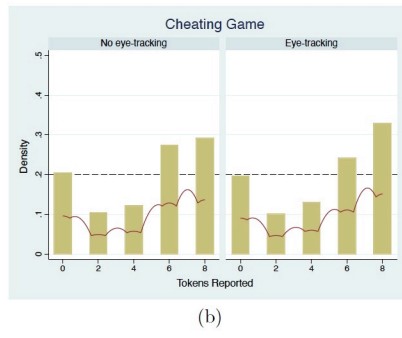

(a)                                                    (b)

**Fig 9. Mean in the average number of tokens reported in 10 periods and distribution comparisons in the number of tokens reported in 10 periods.** Note. Dashed line indicates the expected density of each token occurring, in which a truthful distribution (0.2) [92, 94]. Red lines in (b) present the univariate kernel density estimation.

of failed calibration attempts is correlated with higher measured risk aversion in both games. We then examine potential selection bias into who has difficulty eye-tracking and do not find significant demographic differences between those who do and do not have difficulty calibrating in our sample. We next remove participants who have trouble during the calibration stage of the experiment and find that the significant eye-tracking effect in HL goes away. These results are robust to different thresholds for defining calibration difficulty as explained in section *Results of removing subjects with poor eye-tracking/calibration data*. Then, given that people who have trouble calibrating on the eye-tracker often do not have usable eye-tracking data and are thus excluded from reported results, we instead drop likely excluded participants, those with poor eye-tracking, from our analyses to see if the significant difference in HL goes away, which it does. Finally, we use unobtrusive facial expression analysis software, which does not require calibration, to explore why increased eye-tracking calibration attempts may affect measured risk aversion.

## Poor calibration is correlated with higher measured risk aversion

We first show that the number of failed calibration attempts is correlated with higher measured risk aversion in both risk games. We calculate the number of failed calibration attempts for HL and EG by summing the failed calibration attempts prior to the start of each game. The frequency of the number of failed calibration attempts is displayed in Table D5 in S4 Appendix. Using OLS estimation in Table 5, columns (1)-(6), we find that the number of failed calibration attempts significantly increases the number of safe choices in both risk games by about 0.2, even with the inclusion of the demographic characteristics and the removal of participants with inconsistent preferences. Recall that HL and EG were played by different participants and in different orders, with HL as the third game in Group 1, and EG as the first game in Group 2.

## Demographic characteristics of poor eye-tracking calibrators

We examine the characteristics of subjects across the number of failed calibration attempts for HL and EG to determine if differences in risk aversion are driven by differences in demographic characteristics related to calibration difficulty rather than being caused by failed calibration attempts themselves. Using the Kruskal-Wallis test for means comparison of the cumulative number of calibrations and the average number of calibrations, we find no

**Table 5. Relationship between number of safe choices and cumulative number of calibration attempts prior to game.**

|  | (1) | (2) | (3) | (4) | (5) | (6) |
|---|---|---|---|---|---|---|
|  | Holt and Laury | | | | Eckel and Grossman | |
|  | All Sample | | No Switchers | | | |
| # calibration attempts | 0.185*** | 0.182** | 0.192** | 0.201** | 0.220** | 0.226* |
|  | (0.057) | (0.063) | (0.068) | (0.069) | (0.085) | (0.106) |
| Controls | N | Y | N | Y | N | Y |
| N | 122 | 120 | 99 | 97 | 72 | 71 |
| R-squared | 0.04 | 0.19 | 0.04 | 0.16 | 0.05 | 0.26 |

Note. Coefficients from OLS estimations for the treatment groups only are reported. Standard errors are presented in parentheses and clustered at the session level. No switchers removes subjects with inconsistent preferences. Controls include male, age, year in school indicators, race indicators, and categorical family income variables.

* Statistically significant at 10% level;

** at 5% level;

*** at 1% level.

significant differences across demographic characteristics of gender, age, educational status, race, or income in the cumulative number of calibrations for either HL or EG. These results are available from the authors. There may be demographic characteristics that we did not observe that are correlated with both difficulty calibrating and increased risk aversion, but we do not find any correlations for these first-order possibilities. (In addition to demographic characteristics, calibration difficulties are caused by things like unusually wet eyes, highly reflective glasses, air bubbles under contact lenses, droopy eyelids, small pupils, or lighting conditions [58, 60, 95]).

### Results of removing subjects with poor eye-tracking/calibration data

Given that the number of failed calibration attempts is significantly related to the risk averse behavior in both HL and the EG and does not seem to be correlated with major demographic characteristics that would cause selection bias, we explore the potential solution of simply removing poor calibrators directly and then removing those with poor eye-tracking data over-all, a standard practice in eye-tracking research.

The cumulative number of calibration attempts ranges between 1 and 10 for HL, the only game in which an eye-tracking effect was found. Using the Mann-Whitney test and OLS esti-mation, where standard errors are clustered at the session level, we find that the eye-tracking effect disappears when we exclude subjects who had a cumulative number of calibration attempts greater than 6 in HL ($\sim$ 10 percent of observations), with and without demographic controls included (MW, $p = 0.13$). Table 6, columns (1)-(4), present the OLS results gradually removing people with the most difficulty calibrating (results, not shown, are similar including controls). Just removing the one person with over 10 calibration attempts does not remove the significant eye-tracking result ($\beta = 0.501$, $t = 2.18$), but removing subjects with 9 and 8 calibra-tion attempts reduces the magnitude and significance of the effect ($\beta = 0.441$, $t = 1.88$, and $\beta = 0.439$, $t = 1.83$, respectively). Removing subjects with 7 calibration attempts reduces it further to $\beta = 0.383$, $t = 1.55$. This pattern continues (in results not shown) with the sign of the eye-tracking effect eventually flipping negative when people with more than 2 calibration attempts are removed (N = 138).

Multiple calibrations were attempted when the eye-tracker could not perfectly catch the subject's eyes. Thus, multiple calibration attempts are correlated with low quality eye-tracking data, something that is generally measured as the fraction of frames the eyes were detected

**Table 6. Holt and Laury number of safe choices OLS with universe restrictions.**

|  | (1) | (2) | (3) | (4) | (5) | (6) |
|---|---|---|---|---|---|---|
|  | Cutting by Calibration Attempts | | | | Cutting by data quality % | |
|  | #Cal$\geq$10 | #Cal$\geq$9 | #Cal$\geq$8 | #Cal$\geq$7 | 80% | 85% |
| Eye-Tracking | 0.501** | 0.441* | 0.439* | 0.383 | 0.320 | 0.084 |
|  | (0.230) | (0.234) | (0.240) | (0.247) | (0.211) | (0.189) |
| N | 247 | 243 | 238 | 235 | 205 | 196 |
| R-squared | 0.02 | 0.01 | 0.01 | 0.01 | 0.01 | 0.00 |

Note. Coefficients from OLS estimations are reported. Standard errors are presented in parentheses and clustered at the session level. Columns (5) and (6) use a normal filter which discards the observations that have both eyes have values of 4 or 2 (i.e. (4,4) or (2,2)). Results using a medium and high filter are similarly insignificant, though the magnitudes using the high filter (both eyes have values of 0) are smaller.

* Statistically significant at 10% level;

** at 5% level;

*** at 1% level.

(valid samples) over the total number of recorded frames (total samples) [60, 96, 97]. Dropping low quality data (i.e. lower than an 85 percent threshold according to iMotions) is standard practice in most eye-tracking studies. We first test for an eye-tracking effect in HL with this standard 85 percent threshold cut, which drops 16 percent of the observations. As shown in Table 6, column (6), the eye-tracking effect goes away when these low eye-tracking data quality are removed ($\beta$ = 0.084, $t$ = 0.45). For researchers who want a lower quality threshold, we also tested the effect of removing observations at the 80 percent threshold shown in Table 6 column (5), and found that although the magnitude is larger ($\beta$ = 0.320, $t$ = 1.51), the difference is not significant and is similar to what is found by dropping observations with 7 or more calibration attempts. The results presented in Table 6 columns (5) and (6) assume a normal filter, but results with medium and high filters are similarly insignificant, with magnitudes diminishing as expected, and are available from authors. We also test the EG data to make sure that the null result is robust to dropping calibrators or poor quality data and it is. Similarly, null eye-tracking effects in the other 8 games are also robust to removal of poor calibrators and those with poor quality eye-tracking data (results are available from the authors upon request). The Pearson correlation between calibration attempts greater than 6 and the low eye tracking data quality is 0.055 ($p$ = 0.023), validating that the subjects who experienced difficulties during the calibration stage have the low eye tracking data quality.

## Discussion and conclusion

Our study reveals that using eye-tracking equipment does not affect individual behavior in economic decision making for seven out of eight popular economic games. Participants, on average, behaved the same in both *eye-tracking* and *no eye-tracking conditions*, particularly after implementing standard procedures, such as controlling for the number of participants per session or clustering standard errors at the session level in the Double Auction. Additionally, results from the individual games are in line with previous studies using the same games. In one of our risk preference games, HL, there is a significant difference between the outcomes of the *eye-tracking* and *no eye-tracking* groups. Further exploration into this outcome suggests that subjects who have difficulty with the eye-tracking calibration procedure are driving these differences. The differences in the number of safe choices are still prevalent when we control for demographic differences between subjects who have difficulty calibrating and those who do not, suggesting that the calibration effect is not just driven by selection on observables. One potential explanation is that increased negative emotions from having difficulty calibrating results in the risk averse behavior in HL [98–102].

We began our research with a hypothesis that any eye-tracking effect, if it exists, will be more prevalent in games that are more susceptible to SDB. We feel confident suggesting that SDB emerging from eye-tracking is not a concern in standard economics games. Even the cheating game, where we expected the highest amount of SDB, did not show differences between *eye-tracking* and *no eye-tracking* groups. We speculate, like Andersen et al. [63], that the equipment in these types of eye-tracking experiments might be inconspicuous enough that it is not intrusive to the subjects as the equipment rests on the bottom of the computer monitor and is not physically placed on the subjects' bodies. Additionally, university students are of a generation of individuals who have been continuously immersed in technology from a young age. They might not be as averse to being observed through technology because of the normalized presence of webcams on their phones and computers as well as security cameras in stores and homes.

Although we find no evidence of SDB with eye-tracking, we do find evidence that calibration difficulty can potentially create negative emotions that can affect the risk preferences of bad calibrators. The one game that showed significant differences between the *eye-tracking*

group and *no eye-tracking* group was a risk aversion game, HL. Although testing the effect of calibration on risk aversion was not part of our pre-analysis plan, this correlation holds across two different games (HL and EG) taken by two different sets of people (Group 1 and Group 2), lending credence to the idea that this is a real effect. We suspect that differences in significance between the two games are likely because of the different order that the games were presented, with one of them presented as the first game and the other showing the cumulative effects of calibration difficulties, though we cannot rule out differences in the games themselves or differences between the two subject pools.

Our results suggest that in games that are affected by fear and anxiety, difficulty with calibration may affect the results of risk preference instruments. However, given that calibration difficulty is correlated with eye-tracking data quality, this effect is limited and easy to remedy. It appears that the standard process of removing poor-quality eye-tracking data also removes subjects likely to be affected by calibration problems. Another potential solution to this problem is that researchers could keep track of the number of calibration attempts per subject and should test their results for robustness with and without observations with abnormal numbers of calibration attempts or controlling for the number of calibration attempts. In addition, we found evidence that this calibration effect did not last the full number of rounds, so researchers may also be able to remove the initial data from games with multiple rounds. Finally, as eye-tracking software and hardware get better, this calibration effect should diminish.

Future research is needed to study the potential effects of eye-tracking in other domains. Although this article covers many of the most popular economic games with varying levels of social desirability bias, there are still numerous tasks across social science research that could be examined for a potential eye-tracking effect, particularly with tasks involving risk aversion. Additionally, researchers may be interested in knowing if there is an eye-tracking effect on eye movements themselves. Because observing the change in eye movements due to the eye-tracking requires the use of deception (participants in the control would need to be tracked without their knowledge), this type of study cannot be performed by economics researchers who are prohibited from deceiving. Examining eye-tracking effects on eye movement itself can be important for researchers interested in understanding decision-making processes, and thus we leave such questions to future research.

## Supporting information

**S1 Appendix. Abbreviated instructions.**
(PDF)

**S2 Appendix. Figures dividing multiple round games in half.**
(PDF)

**S3 Appendix. Additional figures for Double Auction.**
(PDF)

**S4 Appendix. Additional tables.**
(PDF)

**S1 Data.**
(ZIP)

## Acknowledgments

We are grateful to Rachael Lanier for research assistance. We thank Glenn Harrison, Samir Huseynov, Ian Krajbich, and Michelle Segovia for helpful comments. This RCT was registered

in the American Economic Association Registry for randomized control trials under trial number AEARCTR-0004174.

## Author Contributions

**Conceptualization:** Jennifer Kee, Melinda Knuth, Joanna N. Lahey, Marco A. Palma.

**Data curation:** Jennifer Kee, Melinda Knuth, Joanna N. Lahey, Marco A. Palma.

**Formal analysis:** Jennifer Kee, Melinda Knuth, Joanna N. Lahey, Marco A. Palma.

**Writing – original draft:** Jennifer Kee, Melinda Knuth, Joanna N. Lahey, Marco A. Palma.

**Writing – review & editing:** Jennifer Kee, Melinda Knuth, Joanna N. Lahey, Marco A. Palma.

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
