## [Decision Letter · Decision Letter 0]

25 May 2021

PONE-D-21-14391

Does Eye-tracking have an effect on economic behavior?

PLOS ONE

Dear Dr. Palma,

Thank you for submitting your manuscript to PLOS ONE. After careful consideration, we feel that it has merit but does not fully meet PLOS ONE’s publication criteria as it currently stands. Therefore, we invite you to submit a revised version of the manuscript that addresses the points raised during the review process.

As you will see below I have two very different reports. The first arises from an economist who did not like the paper very much and the second from a psychologist who liked a lot. Since I am economist myself I can understand (and agree with many of them) the critical view of the first report. 

Personally I feel that the paper is interesting. Yes, it is. But given that the potential audience of this piece of research in a journal like PLOS ONE might be diverse (including economists, game theorists, etc...) I would strongly suggest to revise the paper according to the suggestions provided by Referee #1. Honestly, I feel that making the paper for a wider audience is better.

We look forward to receiving your revised manuscript.

Kind regards,

Pablo Brañas-Garza, PhD Economics

Academic Editor

PLOS ONE

Journal Requirements:

2.Thank you for including your ethics statement: 

"This RCT was registered in the American Economic Association Registry for randomized control trials under trial number AEARCTR-0004174. This study was approved by the IRB, protocol IRB2018-1602.".   

Reviewers' comments:

Reviewer's Responses to Questions

**Comments to the Author**

1. Is the manuscript technically sound, and do the data support the conclusions?

Reviewer #1: Partly

Reviewer #2: Yes

2. Has the statistical analysis been performed appropriately and rigorously? 

Reviewer #1: No

Reviewer #2: Yes

3. Have the authors made all data underlying the findings in their manuscript fully available?

Reviewer #1: No

Reviewer #2: Yes

4. Is the manuscript presented in an intelligible fashion and written in standard English?

Reviewer #1: Yes

Reviewer #2: Yes

5. Review Comments to the Author

Reviewer #1: The authors' self-declared goal is to "investigate whether the use of an eye-tracking device induces changes in the economic behavior in incentivized laboratory experiments" (lines 25-26).

They report statistical findings from a laboratory experiment organized around 8 tasks that are well-known in the experimental-economics literature and two treatments (eye-tracking vs no eye-tracking). The results do not show statistically significant differences in behavior across treatments except for risk attitudes measured through the Holt-and-Laury task. The authors argue that those differences are due to difficulties experienced by subjects in the eye-tracking calibration process.

My main concern with the paper is its lack of focus.

I do not see why a "non-invasive" and practically imperceptible eye-tracking technique would cause any change in behavior. The authors motivate their research by stating that "external validity would be limited to situations in which 27 participants know they are being observed" (lines 27-28). But is it not always the case the experimental participants are being observed and that they are well aware of that? Having one's eye-movements observed is not the same as being "watched" by some observing eyes.

It is very much misleading to refer to "watched by eyes" effects detected by other researchers when subjects saw computer-generated eyespots (or an image of eyes) and acted in a more socially-desirable manner. Note that those eyespots (and eyes) were actually blind. The other main reference using eye-tracking glasses (and provocative swimsuit calendars) is also irrelevant here, not only because it used deception, but because it is not related to economic decision-making.

In spite of the above, let us suppose that eye-tracking might alter behavior. The related experiment should then control for the effect of calibration of the eye-tracking device (given that calibration is necessary for eye- tracking). We would need a baseline treatment with no calibration and no eye-tracking, a treatments with calibration and eye-tracking, and also one with calibration and no eye-tracking. The fourth combination of no calibration and eye-tracking is technically impossible. The paper should be refocused. Instead of putting the experimental tasks in the spotlight, it should pay more attention to the fine details of the eye-tracking process. It should not only report technical details of the device and explain the calibration process and whether all observations were included in the statistical analysis or not, but also control for all those details in the statistical analysis.

Right now, although the paper talks about eye-tracking effects, I find that the data and the statistical results are more about calibration effects.

My remaining concerns are related to the statistical analysis.

* When analyzing acceptance decisions from the ultimatum game, the authors should control for the size of the initial offer.

* Similarly, data were collected over several periods in the public-goods game and the double auction. Both game are known to produce time trends, so the authors should control for that.

* Please avoid the term "marginally significant" and refrain from interpreting "differences" with a p-value larger than 0.05.

* As a robustness test, check the OLS findings with a regression model that is more appropriate for count data (Holt-and-Laury task, Eckel-and-Grossman task).

Reviewer #2: The present work aims to explore whether the use of eye-tracking measures influences the outcome in eight different tasks frequently used in the field of experimental economics (e.g., through an increase in social desirability). From all the tasks, only the Holt and Laury risk assessment task shows differences when control and treatment (eye-tracking) conditions are compared. More specifically, they found that the individuals with a greater number of calibration attempts presented more risk aversion behavior. When removed, there were no differences between conditions. The authors concluded that the use of eye-tracking does not impact the quality of the data obtained in these tasks.

The manuscript is well-written and methodologically sound; besides, the results obtained support the use of a methodology with the potential to make significant contributions in experimental economics. While I recommend it for publication, I have few comments that I include below.

Lines 29 to 33. The authors employed tasks that have several levels of socially desirable choices. I wonder whether they examined differences (for example, in effect size) between tasks with no socially preferred choice and those that have.

Lines 48 to 69. Because earlier studies revealed a short-lived social-desirability effect associated with the use of eye-tracking devices and the authors reviewed such literature in the introduction, an analysis including, for example, two halves of each task could share some light to such effect in the domain of experimental economics.

Lines 610 to 612. Differences in the number of calibration attempts across tasks should be included, even if they are not expected—also, an explanation of why calibration attempts affect the HL but not the others.

Line 626. Could they explain a bit further how the estimation of the emotional states works?

Minor:

Line 71-> Name the Hawthorne effect earlier when including the impact of being observed in other studies so that a reader that is not familiar with the term can fully understand its relevance (for example, between lines 39 to 45).

Typos

Line 284 -> a “T” in theoretically

Line 287-> Frey and Bohnet find(without the final “s”)

6. PLOS authors have the option to publish the peer review history of their article (what does this mean?). If published, this will include your full peer review and any attached files.

Reviewer #1: No

Reviewer #2: No

---

## [Author Response · Author response to Decision Letter 0]

28 Jun 2021

Please see the attached Response to Reviewer file with details about each reviewer comment and our response on how we address each comment. Thank you for your comments and suggestions.

---

## [Decision Letter · Decision Letter 1]

6 Jul 2021

Does Eye-tracking have an effect on economic behavior?

PONE-D-21-14391R1

Dear Dr. Palma,

We’re pleased to inform you that your manuscript has been judged scientifically suitable for publication and will be formally accepted for publication once it meets all outstanding technical requirements.

Kind regards,

Pablo Brañas-Garza, PhD Economics

Academic Editor

PLOS ONE

Additional Editor Comments (optional):

Reviewers' comments:

Reviewer's Responses to Questions

**Comments to the Author**

1. If the authors have adequately addressed your comments raised in a previous round of review and you feel that this manuscript is now acceptable for publication, you may indicate that here to bypass the “Comments to the Author” section, enter your conflict of interest statement in the “Confidential to Editor” section, and submit your "Accept" recommendation.

Reviewer #1: All comments have been addressed

Reviewer #2: All comments have been addressed

2. Is the manuscript technically sound, and do the data support the conclusions?

Reviewer #1: Yes

Reviewer #2: Yes

3. Has the statistical analysis been performed appropriately and rigorously? 

Reviewer #1: Yes

Reviewer #2: Yes

4. Have the authors made all data underlying the findings in their manuscript fully available?

Reviewer #1: No

Reviewer #2: Yes

5. Is the manuscript presented in an intelligible fashion and written in standard English?

Reviewer #1: Yes

Reviewer #2: Yes

6. Review Comments to the Author

Reviewer #1: I have no further comments at this point, and I do now wish to insist on the points raised earlier.

Reviewer #2: Thank you for addressing the comments.

Just a couple of very minor things:

Page 2, line 57-60: The clarification in parenthesis doesn't seem necessary.

Please, check typos and consistency across references (for example, keep all journals' names abbreviated)

7. PLOS authors have the option to publish the peer review history of their article (what does this mean?). If published, this will include your full peer review and any attached files.

Reviewer #1: No

Reviewer #2: No

---

## [Editor Report · Acceptance letter]

19 Jul 2021

PONE-D-21-14391R1 

Does eye-tracking have an effect on economic behavior? 

Dear Dr. Palma:

I'm pleased to inform you that your manuscript has been deemed suitable for publication in PLOS ONE. Congratulations! Your manuscript is now with our production department. 

Kind regards, 

on behalf of

Dr Pablo Brañas-Garza 

Academic Editor

PLOS ONE